# Preliminary Study of Different Treatment Responses between Bevacizumab, Aflibercept and Dexamethasone Implant According to Renal Function in Diabetic Macular Edema Patients

**DOI:** 10.3390/jcm11237047

**Published:** 2022-11-29

**Authors:** Tae Hwan Moon, Gwon Hui Jo, Eoi Jong Seo, Kyung Tae Kim, Eu Jeong Ku, Soon Kil Kwon, Jin Young Kim, Ju Byung Chae, Dong Yoon Kim

**Affiliations:** 1First Eye Clinic, Cheongju 28425, Republic of Korea; 2Department of Ophthalmology, Chungbuk National University Hospital, College of Medicine, Chungbuk National University, Cheongju 28644, Republic of Korea; 3Department of Internal Medicine, Chungbuk National University Hospital, College of Medicine, Chungbuk National University, Cheongju 28644, Republic of Korea; 4Retina Division, The One Seoul Eye Clinic, Seoul 06035, Republic of Korea; 5Top Retina Center, Cheongju 28378, Republic of Korea

**Keywords:** diabetic macular edema, intravitreal anti-VEGF injection, intravitreal dexamethasone implant, renal function

## Abstract

Background: The purpose of this study was to investigate the association between responses to intravitreal bevacizumab injection and renal function in diabetic macular edema (DME) patients. Methods: A retrospective study of the medical records of 104 treatment-naïve DME patients who received intravitreal bevacizumab injection (IVBI) was conducted. Based on the estimated glomerular filtration rate (eGFR, mL/min/1.73 m^2^), the participants were classified into three groups. Intergroup comparisons of the best-corrected visual acuity (BCVA) and central subfield retinal thickness (CST) changes were performed after three-monthly consecutive IVBIs. In the groups with decreased renal function, the response to further treatment with a different drug was investigated. Results: A total of 104 participants were included in the study: 60 participants in the preserved renal function group (eGFR ≥ 60), 25 participants in the moderate chronic kidney disease (CKD) group (30 ≤ eGFR < 60), and 19 participants in the severe CKD group (eGFR < 30). After three-monthly consecutive IVBIs, BCVA (*p* < 0.001) and CST (*p* < 0.001) were significantly improved only in the preserved renal function group. Following further treatment of patients with decreased renal function, the treatment results were significantly better in those who were switched to aflibercept or dexamethasone implant than in those who were maintained on IVBI. Conclusions: From this preliminary study, we observed that renal function might affect the response to IVBI treatment in patients with DME. In the case of a poor response to initial IVBI treatment for DME in patients with moderate to severe CKD, our study supports switching to the aflibercept or dexamethasone implant.

## 1. Introduction

Diabetes mellitus (DM) is one of the most significant public health challenges, and diabetic retinopathy (DR), a critical ocular complication of DM, is one of the major causes of loss of vision in people in the working age-group [1,2]. Diabetic macular edema (DME), characterized by an accumulation of extracellular fluid in the macula as a consequence of the failure of the blood-retinal barrier (BRB), is the most common reason for vision loss in those with DR [3,4]. The prevalence of DME was reported to be 4.2–7.9% in patients with type 1 DM and 1.4–12.8% in those with type 2 DM. However, after 25 years duration of DM, the incidence of DME is approximately 30% in both type 1 and type 2 DM patients [5,6].

Intravitreal injection using anti-vascular endothelial growth factor (VEGF) agents or corticosteroids is the standard of care for DME [4], and various studies have reported good treatment outcomes [7,8,9,10]. However, in some cases, the response to treatment is poor [10]. Since DME is a complication of DM, which is a systemic disease, metabolic control (serum glucose, blood pressure, hyperlipidemia, etc.) could affect the response to treatment in patients with DME. Several studies have investigated various biomarkers related to metabolic control that could possibly predict the response to treatment for DME. These studies reported that some biomarkers, such as glycated hemoglobin (HbA_1C_), blood pressure, serum lipids, and serum creatinine, might be helpful in predicting the response to DME therapy [11,12,13]. However, contrasting results indicating no association between these biomarkers and response to DME therapy were also reported in other studies [14,15]; thus, the outcomes are still debatable.

We previously reported that a significant negative correlation was observed between renal function and subcentral retinal thickness (CST), and suggested that renal function could be used as a biomarker to predict treatment responsiveness in DME patients [16]. Few studies have evaluated the association between the biomarkers of renal function and the functional and anatomic outcomes of anti-VEGF therapy for DME. Therefore, we investigated treatment responsiveness to intravitreal bevacizumab injection (IVBI) for DME according to the renal function, using the estimated glomerular filtration rate (eGFR) as a biomarker of the renal function. 

## 2. Methods

A retrospectively review was conducted on treatment-naïve patients with DME who received IVBI at Chungbuk National University Hospital, Cheongju, South Korea, between January 2014 and June 2020. This study was approved by the institutional review board of Chungbuk National University Hospital (approval number: 2020-12-011) and followed the tenets of the Declaration of Helsinki.

The primary objective of this study was to analyze the response to IVBI therapy for DME according to the renal function of the patients. The secondary objectives were to analyze the difference in the long-term response to different drugs used in the treatment of DME in patients with poor renal function who underwent three-monthly consecutive IVBIs.

We included treatment-naïve DME patients with a CST greater than 300 µm who were initially treated with IVBI. Serologic examinations related to diabetes and kidney function were performed within one month prior to the first IVBI. We excluded patients with media opacities (corneal disease, cataract, vitreous hemorrhage), combined retinal disease, high myopia (>8 diopters), glaucoma, history of intraocular surgery or ocular trauma. And we also excluded DME patients who undergone dialysis due to end-stage renal disease.

### 2.1. Ophthalmic Examinations

During the initial visit, all patients underwent a comprehensive bilateral ophthalmic examination. This included a best-corrected visual acuity (BCVA) using the Snellen chart, slit-lamp examination, measurement of intraocular pressure, fundus examination, and spectral-domain optical coherence tomography (Spectralis; Heidelberg Engineering, Heidelberg, Germany) examination. The CST was defined as the mean retinal thickness in a 1-mm diameter circular zone centered on the fovea. The BCVA results were converted to the LogMAR scale. During each visit, ophthalmic examinations that included BCVA measurement, slit-lamp examination, applanation tonometry, dilated fundus examinations, fundus photography, and SD-OCT were performed.

### 2.2. Laboratory Examinations

Serological tests related to diabetes and kidney function were performed within one month prior to the first IVBI. Venous blood samples were taken from the antecubital vein in the morning after a minimum of 8 h of fasting. Concentrations of HbA_1C_, Blood urea nitrogen (BUN), serum creatinine (Cr), eGFR (mL/min/1.73 m^2^) were measured, also, urine tests were performed. All measurements were performed at the Department of Diagnostic Testing, Chungbuk National University Hospital, using commercially available assays. All laboratory examinations related to diabetes and kidney function were conducted at two- to three-month intervals at the endocrinology clinic of Chungbuk National University Hospital.

### 2.3. Measurement of Treatment Responsiveness

Patients were divided into 3 groups according to their eGFR: preserved renal function group comprised those with eGFR ≥ 60 mL/min/1.73 m^2^, moderate chronic kidney disease (CKD) group comprised those with eGFR between 30 and 59 mL/min/1.73 m^2^, and severe CKD group comprised those with eGFR between 15 and 29 mL/min/1.73 m^2^ (severe CKD Group). All patients were treated with three-monthly consecutive IVBIs. Intergroup comparisons for changes in BCVA and CST were performed after three-monthly consecutive IVBIs. In addition, we analyzed whether there was a difference in therapeutic response to the drug used in further treatments after IVBI in the moderate and severe CKD groups. Through a retrospective chart review, the group was reclassified into a group that maintained bevacizumab after three-monthly consecutive IVBIs, a group that switched to aflibercept, and a group that switched to dexamethasone implant. BCVA and CST obtained at the final visit after further treatment were used in this study, and the differences in the changes in BCVA and CST among the three groups were analyzed.

### 2.4. Statistical Analysis

The Statistical Package for the Social Sciences version 22.0 (SPSS, Inc., Chicago, IL, USA) was used in this study, and *p* < 0.05 was considered as statistically significant. The Kruskal–Wallis test was used to compare continuous variables between groups, and a linear by linear chi-square test was used to compare categorical variables between groups. The Wilcoxon signed-rank test was used to evaluate the difference in parameters between the initial and post-treatment visits within each group.

## 3. Results

A total of 104 treatment-naïve patients with DME were included. Of the 104 patient eyes, 60 (57.7%) were classified as preserved renal function group (eGFR ≥ 60), 25 (24.0%) as the moderate CKD group (30 ≤ eGFR < 60), and 19 (18.3%) as the severe CKD group (eGFR < 30). There were no significant differences in age, sex ratios, the type of DM (type 1 DM, type 2 DM), severity of DR, and refractory errors among the three groups. However, the patients who had poor renal function were more likely to have lower eGFR and HbA1c, longer duration of diabetes, and poorer results of the urine tests (Urine albumin to creatinine ratio and urine microalbumin) and blood tests related to renal function such as BUN and serum creatinine (Table 1). 

### 3.1. Treatment Responsiveness of DME according to Renal Function

After three-monthly consecutive IVBIs, BCVA (*p* < 0.001) and CST (*p* < 0.001) were significantly improved in the preserved renal function group; however, there was no difference in BCVA (*p* = 0.758, *p* = 1.00) and CST (*p* = 0.767, *p* = 0.227) in the moderate and severe CKD groups, respectively. In the intergroup analysis after three-monthly consecutive IVBIs, CST was significantly improved in the preserved renal function group compared with the moderate and severe CKD groups (112.28 ± 148.83 vs. 16.20 ± 159.12 vs. 47.21 ± 128.22, *p* = 0.004). Regarding the severity of DR, we investigated whether there was a difference in the DME treatment outcome according to the severity of DR. Of the total 104 subjects, 50 were Nonproliferative diabetic retinopathy (NPDR) and 54 were Proliferative diabetic retinopathy (PDR) patients. After three IVBIs, there was no significant difference between the two groups (NPDR vs. PDR) in BCVA change (*p* = 0.125) and CST change (*p* = 0.365). In addition, a comparative analysis of treatment results according to DR severity and renal function was performed. We found that there was no significant difference in BCVA change (*p* = 0.067) and CST change (*p* = 0.118) between NPDR and PDR in patients with adequate renal function. In addition, there was no significant difference in treatment outcome according to the severity of DR in the moderate (*p* = 0.156, *p* = 0.531) and severe CKD group (*p* = 0.278, *p* = 0.058). Moreover, patients with preserved renal function showed a better improvement in BCVA; however, the difference between the three groups was not statistically significant in BCVA improvement (−0.14 ± 0.28 vs. −0.07 ± 0.44 vs. −0.01 ± 0.35, *p* = 0.214) (Figure 1).

### 3.2. Response to Further Treatment for DME in Patients with Moderate to Severe CKD

Since there was no significant therapeutic response to three-monthly consecutive IVBIs in patients with moderate to severe CKD, we analyzed whether there was a difference in therapeutic response according to the drug used in further treatments. Twenty-two patients with moderate to severe CKD who received further treatment were included. Ten patients (45.45%) were maintained on IVBI, 5 (22.73%) were switched to aflibercept, and 7 (31.82%) were switched to dexamethasone implant for further treatment. The follow-up period of the further treatment was 8.50 ± 5.30 weeks for the IVBI maintenance group, 6.80 ± 2.1 weeks for the aflibercept switching group, and 10.00 ± 7.96 weeks for the dexamethasone implant switching group. There was a significant improvement in CST in the group that was switched to aflibercept (*p* = 0.042) or dexamethasone implant (*p* = 0.018). In the intergroup analysis, significant differences were observed in the degree of change in BCVA (*p* = 0.043) and CST (*p* = 0.001) between the three groups after further treatment (Figure 2). In the post hoc analysis, the improvement in CST was significantly better in the group that was switched to aflibercept between the two groups that were switched to different drugs (*p* = 0.048), but there was no significant difference in the degree of change in BCVA (*p* = 0.343) (Table 2). 

## 4. Discussion

In this study, we investigated the response to treatment for DME with IVBI, based on the renal function of the patients using the eGFR as a biomarker of the renal function. Good response to IVBI was observed only in the preserved renal function group. In patients with decreased renal function, outcomes of further treatment were significantly better in the groups that were switched to aflibercept or dexamethasone implant than those in whom IVBI was continued as maintenance therapy.

In the Korean medical insurance system, insurance is covered for aflibercept or ranibizumab only in limited cases where: (1) glucose control is well-managed (2) CST is more than 300 µm, and (3) photoreceptor ischemia is not observed. If CST is less than 300 µm, glucose control is poor, or photoreceptor ischemia is observed, it is difficult to consider aflibercept or dexamethasone implant as the first line drug. Therefore, since the number of participants was not sufficient to design a study with patients treated with aflibercept or dexamethasone implant as an initial treatment drug, the study was designed for patients treated with IVBI.

Previous studies have reported that biomarkers of renal function such as BUN, serum creatinine, eGFR, and proteinuria might be associated with anatomical or visual improvement after IVBI therapy in patients with DME [13,16]. To our knowledge, there is only one other study that used eGFR as biomarkers to evaluate their association with the response to DME therapy [17]. Lai et al. reported that patients with an eGFR < 30 had poorer visual improvements than those with normal eGFR, and patients with severe proteinuria showed better anatomical improvement. This is similar to the results of the present study. However, there were some differences in their research design compared with that of our study. Lai’s study compared the treatment outcomes after 1 year of ranibizumab treatment, and urine albumin to creatinine ratio (UACR) was used as a biomarker for proteinuria. In this current study, we compared early treatment response to bevacizumab treatment and also investigated whether switching to aflibercept or dexamethasone implant showed better treatment outcomes in patients with compromised renal function.

Several well-known factors related to CKD can affect the response to DME therapy, including (1) increased vascular hyperpermeability, (2) volume overload, and (3) lower intravascular oncotic pressure. Patients with CKD were found to have elevated serum VEGF, and it could lead to increased vessel permeability by increasing the phosphorylation of tight junction proteins; thus, it is an important mediator of the BRB breakdown [18]. Body fluid status can be associated with the treatment outcome of DME. Both extracellular water and an overhydration status reflect CST in DR patients, overhydration was particularly strongly associated with DME [19]. Decrease in CST with systemic furosemide treatment were reported and we previously reported that after initial dialysis in patients with end-stage renal disease patients, the CST was significantly improved [20,21]. In addition, patients with proteinuria might have low intravascular oncotic pressure, which drives fluid leakage into the extravascular space according to Starling’s rule, resulting in thicker CST [22].

In addition to the well-known factors mentioned above, other factors associated with decreased renal function might affect the response to DME therapy, such as placental growth factor (PlGF) and chronic inflammation. PlGF belongs to the VEGF family and signals directly through VEGF receptor-1 [23]. 

Previous studies reported that the levels of PlGF were elevated in the vitreous and aqueous humor of patients with DR, and PlGF levels are increased in patients with decreased kidney function [24]. Increased PlGF level in DR patient might play an important role in the breakdown of the BRB [25,26]. Inflammation also has an important role in the development of DME. High levels of VEGF increase the expression of the inflammatory intercellular adhesion molecule-1 (ICAM-1), leading to retinal capillary leukostasis, resulting in enhanced vascular permeability and capillary nonperfusion [27]. Therefore, considering multifactorial etiology of DME especially in patients with decreased renal function, treatment responsiveness might be decreased if only the VEGF isoforms are inhibited for DME treatment.

From the results of this preliminary study, we observed that in patients with moderate to severe CKD, further treatment outcomes were significantly better in the groups that were switched to aflibercept or dexamethasone implant than in the group that was maintained on IVBI. aflibercept is a human fusion protein of the IgG Fc region and VEGF-receptor ligand-binding element that inhibits not only VEGF-A but also VEGF-B and PlGF [28]. In addition, corticosteroids with their anti-inflammatory actions block the various steps involved in leukostasis, including downregulation of the selectins and integrins, and also decrease the VEGF synthesis [29,30]. Therefore, in patients with renal dysfunction, who are likely to have high levels of VEGF and PlGF and chronic inflammatory conditions, aflibercept or dexamethasone implant should be considered as the first-line therapy for the treatment of DME.

The strength of our study lies not only in the comparison of the initial treatment responsiveness to DME therapy according to the kidney function of the patients, but also in the reporting of the results of long-term treatment for DME in the group with poor renal function. However, this study has the following limitations: (1) It was a small size, retrospective study and might also have a selection bias. The small sample size is also thought to be related to the big standard deviation of the test result; (2) Since patients were classified by renal function before treatment, the possibility of differences in the rate of decline in renal function in patients during the treatment period was not considered; (3) The number of eyes included in the further treatment is small, and the number of eyes included in the switching groups is relatively small compared with those in the bevacizumab maintenance group; and (4) Different treatment outcomes for DME according to the therapeutic agents in patients with poor renal function were evaluated through non-randomized subjects due to the retrospective study design. In addition, as a cause of this retrospective design, the follow-up period of further treatment was slightly heterogeneous. Therefore, further large-scale, multicenter, prospective studies regarding the association between renal biomarkers and response to DME treatment are necessary.

In conclusion, a good responsiveness to IVBI was observed only in patients of DME with preserved renal function. In patients with moderate to severe CKD, further treatment outcomes were significantly better after switching to aflibercept or dexamethasone implants than with maintaining IVBI therapy. Therefore, eGFR as an indicator of kidney function might be considered a reference biomarker to predict treatment responsiveness in patients with DME. In addition, in the case of a poor response to initial IVBI treatment for DME in patients with moderate to severe CKD, it might be helpful to consider switching to aflibercept or dexamethasone implants.

## Figures and Tables

**Figure 1 jcm-11-07047-f001:**
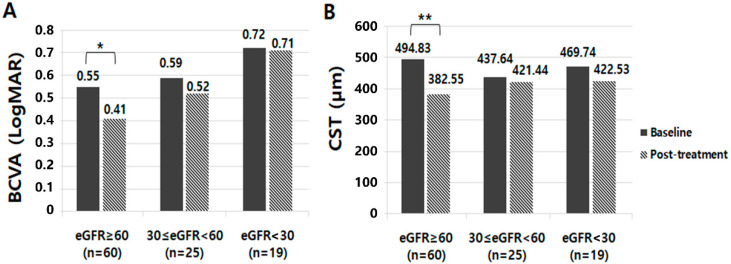
Comparison of the treatment responsiveness of diabetic macular edema after three-monthly consecutive IVBIs according to renal function. (**A**) After three-monthly consecutive IVBIs, BCVA (* *p* < 0.001) were significantly improved in the preserved renal function group; however, there was no difference in BCVA in the moderate and severe CKD groups, respectively (*p* = 0.758, *p* = 1.00). However, in the intergroup analysis, the difference between the three groups in BCVA improvement was not statistically significant (−0.14 ± 0.28 vs. −0.07 ± 0.44 vs. −0.01 ± 0.35, *p* = 0.214). (**B**) After three-monthly consecutive IVBIs, CST (** *p* < 0.001) were significantly improved in the preserved renal function group; however, there was no difference in CST in the moderate and severe CKD groups, respectively (*p* = 0.758, *p* = 1.00). In the intergroup analysis after three-monthly consecutive IVBIs, CST was significantly improved in the preserved renal function group compared with the moderate and severe CKD groups (112.28 ± 148.83 vs. 16.20 ± 159.12 vs. 47.21 ± 128.22, *p* = 0.004).

**Figure 2 jcm-11-07047-f002:**
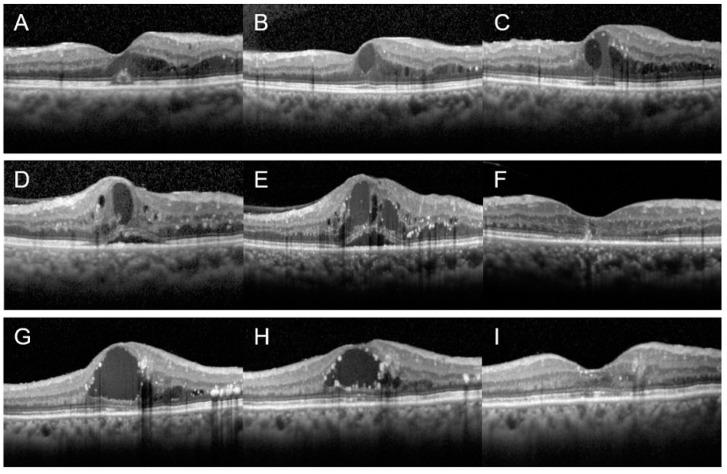
Representative cases of patients with decreased renal function. (**A**–**C**) A case of a 59-year-old woman with moderate CKD who was maintained on IVBI therapy for DME in the left eye. Her estimated GFR was 35.1 mL/min/1.73 m^2^. She was administered three doses of IVBIs; however, her macular edema worsened (**B**) compared with the initial visit (**A**). Bevacizumab injection therapy was continued; however, 4 weeks after further treatment with bevacizumab, the retinal edema was not improved (**C**), and BCVA in the left eye was maintained at 20/70. (**D**–**F**) A case of a 52-year-old women with severe CKD who was switched to aflibercept. Her estimated GFR was 29.5 mL/min/1.73 m^2^. She was administered three doses of IVBIs; however, her macular edema worsened (**E**) compared with the initial visit (**D**). After 8 weeks of aflibercept switching treatment, macular edema was reduced and BCVA also improved from 20/70 to 20/50 (**F**). (**G**–**I**) A case of a 75-year-old woman with severe CKD who was switched to dexamethasone implant. Her estimated GFR was 17 mL/min/1.73 m^2^. After three doses of IVBI, macular edema slightly reduced (**H**) compared with the initial visit (**G**); however, intraretinal fluid was still observed. Four weeks after a single administration of dexamethasone implant, macular edema reduced (**I**) and BCVA also improved from 20/70 to 20/40.

**Table 1 jcm-11-07047-t001:** Comparison of demographic data and baseline characteristics according to renal function.

	eGFR ≥ 60 (n = 60)	30 ≤ eGFR < 60 (n = 25)	eGFR < 30 (n = 19)	*p*-Value
Age (years)	54.70 ± 13.09	57.60 ± 12.73	60.11 ± 8.79	0.219 *
Sex (Male/Female)	32/28	17/8	9/10	0.973 #
Right/Left	33/27	13/12	13/6	0.409 #
Type of Diabetes(1/2)	5/55	2/23	0/19	0.262 #
Diabetes duration (years)	10.36 ± 7.66	13.04 ± 6.93	16.94 ± 9.51	0.008 *
PDR/NPDR	29/31	12/13	8/10	0.795 #
eGFR (mL/min/1.73 m^2^)	110.51 ± 42.70	46.16 ± 7.55	14.56 ± 7.69	<0.001 *
Refractive error (S.E.)	−0.91 ± 2.06	−1.65 ± 2.54	−0.54 ± 1.86	0.214 *
HbA_1C_ (%)	8.26 ± 2.15	8.11 ± 1.75	6.74 ± 0.92	0.045 *
Urine albumin to creatinine ratio (mg/g)	416.35 ± 1219.49	2855.89 ± 3095.72	3358.78 ± 2729.46	<0.001 *
Urine microalbumin (μg/mgCr)	382.45 ± 934.05	1718.22 ± 1321.17	2896.26 ± 2781.10	<0.001 *
BUN (mg/dL)	15.72 ± 4.19	25.56 ± 7.36	51.26 ± 20.43	<0.001 *
Serum creatinine (mg/dL)	0.75 ± 0.20	1.56 ± 0.29	4.68 ± 2.59	<0.001 *
Proteinuria/Normal	28/21	21/1	15/1	<0.001 #

eGFR, estimated glomerular filtration rate; PDR, Proliferative diabetic retinopathy; NPDR, Non proliferative diabetic retinopathy; S.E., Spherical equivalent; HbA_1C_, Glycosylated hemoglobin; BUN, Blood urea nitrogen. *: Kruskall–Wallis test; #: linear by linear association chi square test.

**Table 2 jcm-11-07047-t002:** Comparison of further treatment responsiveness in diabetic macular edema patients with moderate to severe CKD.

	Maintain Bevacizumab (n = 10)	Switch to Aflibercept (n = 5)	Switch to Dexamethasone Implant (n = 7)	*p*-Value
BCVA change (LogMAR)	0.15 ± 0.36	−0.29 ± 0.22	−0.17 ± 0.41	0.043 *
CST change (µm)	32.30 ± 63.87	312.20 ± 51.76	193.86 ± 101.33	0.001 *

CKD, chronic kidney disease; BCVA, best corrected visual acuity; CST, Central subfield thickness. *: Kruskall–Wallis test.

## Data Availability

The datasets used and/or analyzed during the current study are available from the corresponding authors on reasonable request.

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
