# Peer review of "Preliminary Study of Different Treatment Responses between Bevacizumab, Aflibercept and Dexamethasone Implant According to Renal Function in Diabetic Macular Edema Patients"

_jcm, 2022, doi:10.3390/jcm11237047_

Round 1

Reviewer 1 Report

The present manuscript discussed the possible use of aflibercept and dexamethasone in diabetic macular edema patients with renal impairment, in comparison to the intravitreal bevacizumab injection. Overall, this is original study, novel findings have been obtained, the manuscript is generally well written and data are well described. However, there are some concerns that need to be clarified.

Major concern is the sample size. This is also stated in the Limitations of the study. However, the power analysis has to be performed in order to estimate the minimum sample size required for an experiment, i.e. to ascertain if 19 patients with severe chronic kidney disease, 5 patients on aflibercept therapy and 7 patients on dexamethasone therapy are sufficient to draw any reliable conclusion. If it is not sufficient, it might be some preliminary study that only gives directions for future investigations; however, it needs to be clear from the title and the abstract.

Standard deviations are also very big for CST measurements (112.28±148.83 vs. 16.20±159.12 vs. 47.21±128.22) - it should be checked again.

The number of type 1 diabetes patients is very small. Why did you decide to include them in the study, since there are only two (out of 44) T1DM patients among the patients with renal dysfunction?

Besides, why is only aflibercept in the title and not dexamethasone as well? How long was the treatment with these two drugs after switching from IVBI, i.e. after which period were BCVA and CST determined in these patients? It should be clear in the text.

Other concerns are also listed below.

There are some technical issues in the manuscript. For example, the references should be stated before full stop, and not after.

Abstract: Define IVBI where it was first time mentioned (line 20 instead of line 23). I find ‘104 eyes’ redundant, ‘104 patients’ is sufficient.

Line 72: add in the parentheses that it is an approval number.

Line 105: The title of this section should be different – study groups or experimental protocol, this is the title for ‘Results’ section (line 140).

Line 136: please define what kind of ‘characteristic’?

Line 147: the sentence not finished!

Line 149: NPRD is first time mentioned in the main text and it should be defined.

Lines 156-157: “…showed a better improvement in BCVA improvement” - it needs to be corrected.

Line 161: please be specific that it refers to IVBI.

Lines 163-172 and 190-204: are these parts of figure legends? If so, font should be changed and it should be more clear.

Lines 234-236: font should be corrected.

Author Response

<Response to Reviewer 1>

The present manuscript discussed the possible use of aflibercept and dexamethasone in diabetic macular edema patients with renal impairment, in comparison to the intravitreal bevacizumab injection. Overall, this is original study, novel findings have been obtained, the manuscript is generally well written and data are well described. However, there are some concerns that need to be clarified.

Major concern is the sample size. This is also stated in the Limitations of the study. However, the power analysis has to be performed in order to estimate the minimum sample size required for an experiment, i.e. to ascertain if 19 patients with severe chronic kidney disease, 5 patients on aflibercept therapy and 7 patients on dexamethasone therapy are sufficient to draw any reliable conclusion. If it is not sufficient, it might be some preliminary study that only gives directions for future investigations; however, it needs to be clear from the title and the abstract.

  • Thank you for your valuable comment. We changed the title of the article to “Preliminary study of different treatment responses between bevacizumab, aflibercept and dexamethasone implant according to renal function in diabetic macular edema patients” Also, this article is a preliminary study and is described in the conclusion of the abstract as follows;

From

“Conclusions: Renal function might affect the response to IVBI treatment in patients with DME”

To

“Conclusions: From this preliminary study, we could know that renal function might affect the response to IVBI treatment in patients with DME.”

Standard deviations are also very big for CST measurements (112.28±148.83 vs. 16.20±159.12 vs. 47.21±128.22) - it should be checked again.

  • The statistical results were reviewed, and the presently described results are correct. The standard deviation was large because it included all cases in which CST significantly decreased after treatment or increased despite treatment. Also, the small sample is also a reason for the large standard deviation Regarding this, it was additionally described in the limitations section as follows;

Lines 276-277

 “The small sample size is also thought to be related to the big standard deviation of the test result.”

The number of type 1 diabetes patients is very small. Why did you decide to include them in the study,

since there are only two (out of 44) T1DM patients among the patients with renal dysfunction?

  • In order to include as many participants as possible during the study period, type1 DM patients were not specifically excluded.

Besides, why is only aflibercept in the title and not dexamethasone as well? How long was the treatment with these two drugs after switching from IVBI, i.e. after which period were BCVA and CST determined in these patients? It should be clear in the text.

  • As mentioned above, the title has been changed to reflect what you pointed out. For the results of further treatment according to the drug switching, the test results measured at the next visit after the treatment switch were analyzed. As this is a retrospective study, the follow-up period is slightly heterogeneous. The follow-up period was additionally described in the text (lines 183-186). Regarding the heterogenous follow-up period, it was additionally described in the limitation session (lines 284-285).

Lines 183-186

 “The follow-up period of the further treatment was 8.50±5.30 weeks for the IVBI maintenance group, 6.80±2.1 weeks for the aflibercept switching group, and 10.00±7.96 weeks for the dexamethasone implant switching group.”

Lines 284-285

“In addition, as a cause of this retrospective design, the follow-up period of further treatment was slightly heterogeneous”

Other concerns are also listed below

There are some technical issues in the manuscript. For example, the references should be stated before full stop, and not after.

  • The location of the reference was changed before the full stop of the sentence.

Abstract: Define IVBI where it was first time mentioned (line 20 instead of line 23).

  • As you pointed out, we modified it as follows (line 19-20);

Lines 19-20

“A retrospective study of the medical records of 104 treatment-naïve DME patients who received intravitreal bevacizumab injection (IVBI) was conducted.”

I find ‘104 eyes’ redundant, ‘104 patients’ is sufficient.

.

  • The redundant content '104 eyes' has been deleted.

Line 72: add in the parentheses that it is an approval number.

  • We have added a note to indicate that it is an IRB approval number (line 73).

Line 73

(approval number: 2020-12-011)

Line 105: The title of this section should be different – study groups or experimental protocol, this is the title for ‘Results’ section (line 140).

  • As you pointed out, we changed the title of the session from " Treatment responsiveness to the intervention " to "Measurement of treatment responsiveness". (line 106)

Line 106

“2.3. Measurement of treatment responsiveness”

Line 136: please define what kind of ‘characteristic’?

  • The title of table1 has been changed from " Comparison of characteristic according to renal function " to " Comparison of demographic data and baseline characteristics according to renal function " (line 137)

Line 137

“Table 1. Comparison of demographic data and baseline characteristics according to renal function”

Line 147: the sentence not finished!

  • The enter key was entered incorrectly, and it has been corrected

Line 149: NPRD is first time mentioned in the main text and it should be defined.

  • Since NPDR and PDR are mentioned first, the full term of the abbreviation has been defined. (line 151-153)

Lines 151-153

Of the total 104 subjects, 50 were Nonproliferative diabetic retinopathy (NPDR) and 54 were Proliferative diabetic retinopathy (PDR) patients.

Lines 156-157: “…showed a better improvement in BCVA improvement” - it needs to be corrected.

  • As you pointed out, we modified it as follows: (lines 160-161)

From

 “Moreover, patients with preserved renal function showed a better improvement in BCVA improvement”

To

“Moreover, patients with preserved renal function showed a better improvement in BCVA.”

Line 161: please be specific that it refers to IVBI.

  • The caption of figure 1 has been modified as follows.

From

“Comparison of the treatment responsiveness of diabetic macular edema according to renal function”

To

“Comparison of the treatment responsiveness of diabetic macular edema after 3 monthly consecutive IVBIs according to renal function”

Lines 163-172 and 190-204: are these parts of figure legends? If so, font should be changed and it should be more clear.

  • The font of figure legends has been modified differently from the main text.

Lines 234-236: font should be corrected.

  • The font has been modified in the same format as the main text.

Reviewer 2 Report

1.Line 147 ends with an unfinished sentence.

2.The format of line 234-236 needs to be changed.

3.The classification of this experiment was based on the initial eGFR, but it did not mention whether the eGFR changed during the treatment and after 3 months. Because the results and analysis in this paper are based on this grouping, the addition of this result will make the conclusion more reliable.

Author Response

<Response to Reviewer 2>

  1. Line 147 ends with an unfinished sentence.
  • The enter key was entered incorrectly, and it has been corrected
  1. The format of line 234-236 needs to be changed.
  • The font has been modified in the same format as the main text.
  1. The classification of this experiment was based on the initial eGFR, but it did not mention whether the eGFR changed during the treatment and after 3 months. Because the results and analysis in this paper are based on this grouping, the addition of this result will make the conclusion more reliable
  •   Thank you for your valuable comment. This study retrospectively used laboratory data conducted in the Department of Endocrinology. Therefore, in some cases, the eGFR level after 3 months of injection treatment could not be obtained. Therefore, we could not know the eGFR change during treatment. This is described in the limitation session as follows; (lines 278-280)

Lines 278-280

“(2) Since patients were classified by renal function before treatment, the possibility of differences in the rate of decline in renal function in patients during the treatment period was not considered.”

Reviewer 3 Report

This is an interesting study which compared the efficacy of the DME treatment with intravitreal bevacizumab injection and the renal function in patients with both times of DM. However it needs a major revision before further consoderation of process. The main concern is the design of the study ie. why the authors chose to use the bevacizumab which is the off label therapy instead of aflibercept or ranibizumab ? The costs of therapy is not a good reason especially if the switch to aflibercept was prooved by authors to have better results in terms of BCVA and CST values. Most of the DME patients have impaired renal function and according to the previously published studies both afliberecpt and dexamethasone had very good outcomes. You should address this issue in the introduction, discussion and limitation sections. Do you think that the different size of molecules may play any role in the response to the treatment according to renal function ?

Author Response

<Response to Reviewer 3>

This is an interesting study which compared the efficacy of the DME treatment with intravitreal bevacizumab injection and the renal function in patients with both times of DM. However it needs a major revision before further consoderation of process.

The main concern is the design of the study ie. why the authors chose to use the bevacizumab which is the off label therapy instead of aflibercept or ranibizumab ? The costs of therapy is not a good reason especially if the switch to aflibercept was prooved by authors to have better results in terms of BCVA and CST values.

  • Thank you for your valuable comment. Patients with normal renal function who obtained good results in BCVA and CST ​​through IVBI did not switch to aflibercept or ranibizumab, and switching was limited to the group in which there was no significant improvement in BCVA and CST. There are several reasons why we started Initial Treatment with IVBI in the study design. In the Korean medical insurance system, insurance is covered for aflibercept or ranibizumab only in limited cases where (1) glucose control is well, (2) CST is more than 300um, and (3) photoreceptor ischemia is not observed. If CST is less than 300uM, glucose control is poor, or photoreceptor ischemia is observed, it is difficult to consider aflibercept or dexamethasone implant as the first line drug. Therefore, since the number of participants was not sufficient to design a study with patients treated with aflibercept or dexamethasone implant as an initial treatment drug, the study was designed for patients treated with IVBI. A follow-up study in the renal impairment group, which had poor initial IVBI treatment results, will be conducted using aflibercept or dexamethasone implant as a primary treatment choice.

Most of the DME patients have impaired renal function and according to the previously published studies both afliberecpt and dexamethasone had very good outcomes. You should address this issue in the introduction, discussion and limitation sections. Do you think that the different size of molecules may play any role in the response to the treatment according to renal function ?

  • As we know, few studies have analyzed differences in treatment response according to renal function in DME. As you commented, there are many reports of aflibercept and dexamethasone implant showing good responses to DME treatment. In this study, we analyzed treatment response of DME according to the renal function. That could be a unique feature of our clinical study.  And the difference in treatment response is thought to be due to the fact that aflibercept blocks PIGF and dexamethasone shows a therapeutic effect through a different mechanism of reducing inflammation, rather than molecular size. Related contents are described in discussion session. (lines 265-272)

Lines 265-272

“Aflibercept is a human fusion protein of the IgG Fc region and VEGF-receptor ligand-binding element that inhibits not only VEGF-A but also VEGF-B and PlGF. [28]. In addition, corticosteroids with their anti-inflammatory actions block the various steps involved in leukostasis, including downregulation of the selectins and integrins, and also decrease the VEGF synthesis. [29,30]. Therefore, in patients with renal dysfunction, who are likely to have high levels of VEGF and PlGF and chronic inflammatory conditions, aflibercept or dexamethasone implant should be considered as the first-line therapy for the treatment of DME.”

Reviewer 4 Report

This study evaluated the response of DME to anti VEGF based on different level of eGFR, excluding renal failure cases. Please see my comments below:

1. Result section: some sentences should be more clear in the variable they are pointing. For example in the following sentences: "However, in the intergroup analysis, the difference between the three groups was not statistically significant (-0.14±0.28 167 vs. -0.07±0.44 vs. -0.01±0.35, p=0.214)." the difference between what in three groups were insignificant?

or this one " there was no difference in the moderate and severe CKD groups, " do you mean no difference in baseline CST? 2. figure captions should be more complete. what appears in the text for describing the photos seems to have the wrong labeling. The follow up timing should be mentioned for each photo as well. 3. UACR stands for what ? is it urine albumin creatinine ratio. It should be mentioned in the text. 4. In result section, follow up times in weeks/months are not mentioned for different subgroups. for example in the section with title number of 3.2, it is not clear how long after treatment switch, the patients were followed. The follow up times may be heterogeneous, but it is good to mention the average and range, SD.

Author Response

<Response to Reviewer 4>

This study evaluated the response of DME to anti VEGF based on different level of eGFR, excluding renal failure cases. Please see my comments below:

  1. Result section: some sentences should be more clear in the variable they are pointing. For example in the following sentences: "However, in the intergroup analysis, the difference between the three groups was not statistically significant (-0.14±0.28 167 vs. -0.07±0.44 vs. -0.01±0.35, p=0.214)." the difference between what in three groups were insignificant?
  • Thank you for your valuable comments. The part you pointed out is the result of the intergroup analysis of BCVA improvement after IVBI, and this has been described in detail as follows; (lines 161-163)

Lines 161-163

“however, the difference between the three groups was not statistically significant in BCVA improvement (-0.14±0.28 vs. -0.07±0.44 vs. -0.01±0.35, p=0.214) (Fig 1).”

or this one " there was no difference in the moderate and severe CKD groups, " do you mean no difference in baseline CST?

  • It was further described that there was no significant difference in CST. (lines 173-174)

Lines 173-174

“however, there was no difference in CST in the moderate and severe CKD groups, respectively (p=0.758, p=1.00).”

  1. figure captions should be more complete. what appears in the text for describing the photos seems to have the wrong labeling. The follow up timing should be mentioned for each photo as well.
  •  Incorrect captions have been corrected (A to I), and the follow-up period for each patient has been additionally described. (lines 197-209)

Lines 197-209

“(A-C) A case of a 59-year-old woman with moderate CKD who was maintained on IVBI therapy for DME in the left eye. Her estimated GFR was 35.1 mL/min/1.73 m2. She was administered three doses of IVBIs; how-ever, her macular edema worsened (B) compared with the initial visit (A). Bevacizumab injection therapy was continued; however, 4 weeks after further treatment with bevacizumab, the retinal edema was not improved (C), and BCVA in the left eye was maintained at 20/70. (D-F) A case of a 52-year old women with severe CKD who was switched to aflibercept. Her estimated GFR was 29.5 mL/min/1.73 m2. She was administered three doses of IVBIs; however, her macular edema worsened (E) compared with the initial visit (D). After 8 weeks of aflibercept switching treatment, macular edema was reduced and BCVA also improved from 20/70 to 20/50 (F). (G-I) A case of a 75-year old woman with severe CKD who was switched to dexamethasone implant. Her estimated GFR was 17 mL/min/1.73 m2. After three doses of IVBI, macular edema slightly reduced (H) compared with the initial visit (G); however, intraretinal fluid was still observed. Four weeks after a single administration of dexamethasone implant, macular edema reduced(I) and BCVA also improved from 20/70 to 20/40.”

  1. UACR stands for what ? is it urine albumin creatinine ratio. It should be mentioned in the text.
  • UACR is described in full words as follows; (Lines 132-136)

Lines 132-136

“However, the patients who had poor renal function were more likely to have lower eGFR and HbA1c, longer duration of diabetes, and poorer results of the urine tests (Urine albumin to creatinine ratio and urine microalbumin) and blood tests related to renal function such as BUN and serum creatinine (Table 1).”

  1. In result section, follow up times in weeks/months are not mentioned for different subgroups. for example in the section with title number of 3.2, it is not clear how long after treatment switch, the patients were followed. The follow up times may be heterogeneous, but it is good to mention the average and range, SD.
  • As you pointed out, we have additionally described the follow-up period for Further Treatment (line 183-186). Follow-up periods are, as expected, slightly heterogeneous. Regarding the heterogeneous follow-up period, it was additionally described in the limitation session (line 284-285).

Lines 183-186

 “The follow-up period of the further treatment was 8.50±5.30 weeks for the IVBI maintenance group, 6.80±2.1 weeks for the aflibercept switching group, and 10.00±7.96 weeks for the dexamethasone implant switching group.”

Lines 284-285

“In addition, as a cause of this retrospective design, the follow-up period of further treatment was slightly heterogenous”

Round 2

Reviewer 1 Report

Manuscript is improved .

Author Response

Manuscript is improved 

  • Thank you again for your valuable comments for improving the manuscript.

Reviewer 3 Report

However, the present version is improved , it still needs a major revision. Please add to the introduction or discussion the following paragraph . In the Korean medical insurance system, insurance is covered for aflibercept or ranibizumab only in limited cases where (1) glucose control is well, (2) CST is more than 300um, and (3) photoreceptor ischemia is not observed. If CST is less than 300uM, glucose control is poor, or photoreceptor ischemia is observed, it is difficult to consider aflibercept or dexamethasone implant as the first line drug. Therefore, since the number of participants was not sufficient to design a study with patients treated with aflibercept or dexamethasone implant as an initial treatment drug, the study was designed for patients treated with IVBI. 

Author Response

However, the present version is improved , it still needs a major revision. Please add to the introduction or discussion the following paragraph . In the Korean medical insurance system, insurance is covered for aflibercept or ranibizumab only in limited cases where (1) glucose control is well, (2) CST is more than 300um, and (3) photoreceptor ischemia is not observed. If CST is less than 300uM, glucose control is poor, or photoreceptor ischemia is observed, it is difficult to consider aflibercept or dexamethasone implant as the first line drug. Therefore, since the number of participants was not sufficient to design a study with patients treated with aflibercept or dexamethasone implant as an initial treatment drug, the study was designed for patients treated with IVBI. 

  • we added the contents to the discussion session(lines 221-228).
    Thank you for your valuable comment.